# One-Step Hydrothermal Synthesis of Highly Fluorescent MoS_2_ Quantum Dots for Lead Ion Detection in Aqueous Solutions

**DOI:** 10.3390/nano12193329

**Published:** 2022-09-24

**Authors:** Luogang Xie, Yang Yang, Gaoshang Gong, Shiquan Feng, Dewei Liu

**Affiliations:** School of Physics and Electronic Engineering, Henan Key Laboratory of Magnetoelectronic Information Functional Materials, Zhengzhou University of Light Industry, Zhengzhou 450002, China

**Keywords:** molybdenum disulfide, quantum dots, lead ion, fluorescence probe

## Abstract

Lead ions in water are harmful to human health and ecosystems because of their high toxicity and nondegradability. It is important to explore effective fluorescence probes for Pb^2+^ detection. In this work, surface-functionalized molybdenum disulfide quantum dots (MoS_2_ QDs) were prepared using a hydrothermal method, and ammonium tetrathiomolybdate and glutathione were used as precursors. The photoluminescence quantum yield of MoS_2_ QDs can be improved to 20.4%, which is higher than that for MoS_2_ QDs reported in current research. The as-prepared MoS_2_ QDs demonstrate high selectivity and sensitivity for Pb^2+^ ions, and the limit of detection is 0.056 μM. The photoluminescence decay dynamics for MoS_2_ QDs in the presence of Pb^2+^ ions in different concentrations indicate that the fluorescence quenching originated from nonradiative electron transfer from excited MoS_2_ QDs to the Pb^2+^ ion. The prepared MoS_2_ QDs have great prospect and are expected to become a good method for lead ion detection.

## 1. Introduction

Heavy metal contamination in the environment has become an urgent problem to solve because of its threat to human health and ecosystems [1,2]. In industry and agricultural fields, heavy metal ions (Pb^2+^, Hg^2+^, Cd^2+^, etc.) are widely used. If not handled properly, these ions can leak into water circulation systems and further contaminate our drinking water and food, ultimately enriching in the human body and changing protein structure to cause a series of diseases. Lead pollutants are considered to be one of the most dangerous contaminants, which exhibit high toxicity and nondegradability [3,4]. So far, atomic absorption spectrometry, electrochemical technique, inductively coupled plasma mass spectrometry, among others, have been developed to detect heavy metal ions [5,6,7]. However, their application is often limited due to complicated sample pretreatment, long analysis time, and expensive equipment. To overcome these shortcomings, an approach of optical detection based on fluorescence analysis has emerged for its high sensitivity and selectivity. Hence, to detect Pb^2+^ ions, it is urgent to develop an ecofriendly fluorescent material with high sensitivity and selectivity.

Owing to quantum confinement effect and edge effect, semiconductor quantum dots (QDs) possess unique photophysical properties and can be used as a fluorescence probe for ion detection. Fluorescence probes for Pb^2+^ based on QDs have gained much interest. Mn-doped ZnS QDs, ZnSeS/Cu:ZnS/ZnS core/shell/shell QDs, and carbon dots have been developed for Pb^2+^ detection [8,9,10]. However, the inorganic quantum dots are imperfect for their toxicity and multistep synthetic approach. Although carbon dots have created great focus for their facile synthesis and low toxicity, low quantum yield limits their application. In recent years, two-dimensional transition metal dichalcogenides such as MoS_2_ and WSe_2_, and especially their quantum dots, have received much attention in sensing applications and optoelectronic devices [11,12,13,14,15,16,17]. MoS_2_ QDs with large surface-to-volume ratio and abundant active edge sites can be used as a photoluminescence-sensing platform [18,19,20,21]. Different from traditional quantum dots (CdSe, CdTe, etc.) with harmful elements, MoS_2_ QDs are water-soluble and nontoxic. Hence, MoS_2_ QDs have received extensive attention in bioimaging, sensing and photodynamic therapy [22,23]. MoS_2_ QDs have been used as a sensor to detect nitro explosives, hyaluronidase and hydrogen peroxide, along with glucose and other biomolecules [18,19,20,24,25,26]. Additionally, there are some reports on the metal ion sensor using fluorescent MoS_2_ QDs. Cysteine-functionalized MoS_2_ QDs have been used for sensing Al^3+^ and Fe^3+^ metal ions [27]. Based on the fluorescence turn-off effect, MoS_2_ QDs can be used as a sensor for Fe^3+^ detection [28]. However, the fluorescence quantum yield of these MoS_2_ QDs reported by previous research is less than 10%, and the interaction of metal ions with MoS_2_ QDs is not well studied. To obtain a high fluorescence MoS_2_ QDs, a widely used bottom-up method of hydrothermal synthesis has been improved with different molybdenum and sulfur sources, which is simple, ecofriendly and easy to operate. In the hydrothermal method, MoS_2_ QDs were obtained by the Xian group; they successfully adopted ammonium tetrathiomolybdate [(NH_4_)_2_MoS_4_] and hydrazine hydrate as precursor and reducing agent, respectively [24]. Zhang et al. have also synthesized MoS_2_ QDs using this method, with sodium molybdate (Na_2_MoO_4_·2H_2_O) and glutathione (GSH) serving as molybdenum and sulfur sources [27]. Although the photoluminescence quantum yield of MoS_2_ QDs can be increased to 6% in this method, it is challenging to produce large quantities of bright MoS_2_ QDs by using more appropriate Mo and S sources.

Herein, we explore high fluorescent MoS_2_ QDs in a one-step hydrothermal method using (NH_4_)_2_MoS_4_ and GSH as Mo and S sources. On the one hand, GSH was used as a reductant to reduce (NH_4_)_2_MoS_4_; on the other hand, GSH, as a passivation agent, could eliminate the surface defects of MoS_2_ QDs to enhance the fluorescence. If we introduced the Pb^2+^ ions into the resultant MoS_2_ QDs aqueous solution, quenched fluorescence was observed. We further acquired the linear dependence of fluorescence intensity on Pb^2+^ ion concentration in a certain range. The limit of detection (LOD) is 0.056 μM, which is below the acceptable limit given by United States Environmental Protection Agency. We further study the photoluminescence decay dynamics of MoS_2_ QDs with increasing the concentration of Pb^2+^ ion. With a detailed exciton dynamics study, we found that the fluorescence quenching originated from the electron transfer from MoS_2_ QDs to Pb^2+^ ions. As a result, MoS_2_ QDs prepared in this work can be used as a sensing probe to detect Pb^2+^ ions in water with high sensitivity and selectivity. These results are important to understanding the sensing mechanism of MoS_2_ QDs to metal ions.

## 2. Materials and Methods

### 2.1. Chemicals

All chemicals were of analytical grade without further purification. (NH_4_)_2_MoS_4_ (J&K Chemical Ltd., Shanghai, China) and GSH (Aladdin Industrial Co., Ltd., Shanghai, China) were used as precursor and reductant respectively. The metal analytes used in this study were CaCl_2_, MgCl_2_·6H_2_O, MnCl_2_·4H_2_O, PbCl_2_, AlCl_3_, SnCl_2_, FeCl_2_·4H_2_O, CuCl_2_, Fe(NO_3_)_3_·9H_2_O, HgCl_2_, BaBr_2_, ZnCl_2_, CsCl, KCl and AgNO_3_. In addition, HCl (0.1 M) and NaOH (0.1 M) solution was prepared to adjust pH. Quinine sulfate (0.05 M) was chosen as a standard sample to calculate the photoluminescence quantum yield of MoS_2_ QDs in a reference method. Ultrapure water was used throughout the experiment.

### 2.2. Preparation of MoS_2_ QDs

(NH_4_)_2_MoS_4_ and GSH were chosen as Mo and S sources to synthesis MoS_2_ QDs through a pot hydrothermal process. Briefly, (NH_4_)_2_MoS_4_ (65 mg) was dispersed in ultrapure water (25 mL), sonicated for 10 min until the (NH_4_)_2_MoS_4_ was fully dissolved. Afterward, 0.1 M HCl was used to adjust the pH of solution to 6.5. In addition, GSH (308 mg) was dissolved in ultrapure water (50 mL), and subsequently added into the (NH_4_)_2_MoS_4_ aqueous solution prepared in the previous step. The mixture was further sonicated for 10 min and heated at 200 °C in a Teflon-lined autoclave (100 mL) for 24 h. The solution was then naturally cooled to room temperature. The mixture was centrifuged for 10 min at 10,000 rpm to collect the suspension, subsequently purified by a 0.22 µm microporous membrane. The target MoS_2_ QDs aqueous solution was stored at 4 °C for further characterization.

### 2.3. Characterization of MoS_2_ QDs

Commercial experimental instruments including a UV–Vis–NIR spectrophotometer (Cary 5000, Agilent, Santa Clara, CA, USA) and fluorescence spectrometer (PerkinElmer LS 55, Waltham, MA, USA) were used to measure the absorption and photoluminescence spectra, respectively. Time-resolved photoluminescence (TRPL) measurements were conducted with a home-built fluorescence lifetime setup (Picoharp300, Picoquant, Berlin, Germany) based on a time-correlated single photon counting (TCSPC) module. A pulsed excitation source (350 nm with repetition frequency of 13 MHz) was used to excite the samples.

The particle diameters and monodispersity of MoS_2_ QDs were characterized by high resolution transmission electron microscopy (HRTEM) (JEM-2010, JEOL, Tokyo, Japan). The aqueous solution of appropriate concentration was dripped onto a carbon-coated copper grid, and the aqueous solvent was dried by nitrogen flow at ambient temperature.

The monolayer nature of MoS_2_ QDs was checked by atomic force microscopy (AFM, Solver-P47H, NT-MDT, Moscow, Russia). The sample for AFM measurement was acquired by spin coating diluted MoS_2_ aqueous solution on a mica wafer.

Elemental analysis was conducted using X-ray photoelectron spectroscopy (XPS) (PHI 5000 VersaProbeIII, Japan). The binding energy calibration was performed using C 1s of 284.6 eV as standard peak energy.

The surface functional groups of MoS_2_ QDs were identified by Fourier transform infrared spectroscopy (FTIR) spectra (Perkin-Elmer spectrometer, Spectrum One B, Waltham, MA, USA), and the MoS_2_ QDs powder was pressed into a tablet with KBr.

### 2.4. Procedures for Detection of Metal Ions

For the selectivity study of MoS_2_ QDs, an aqueous solution of metal ions including Ca^2+^, Mg^2+^, Mn^2+^, Pb^2+^, Al^3+^, Sn^2+^, Fe^2+^, Cu^2+^, Fe^3+^, Hg^2+^, Ba^2+^, Zn^2+^, Cs^2+^, K^+^ and Ag^+^ was prepared with concentration of 25 μM. For the sensitivity study, Pb^2+^ ion solution was prepared with concentrations of 120 μM, 100 μM, 90 μM, 80 μM, 70 μM, 60 μM, 50 μM, 45 μM, 40 μM, 35 μM, 30 μM, 25 μM, 20 μM, 15 μM, 10 μM and 5 μM. Then, the various metal ions were added into the corresponding MoS_2_ QDs solution to measure the fluorescent intensity by using the fluorescence spectrometer.

## 3. Results

The MoS_2_ QDs were first characterized by TEM. The MoS_2_ QDs are monodispersed and homogeneous with an average diameter of 4.4 ± 0.2 nm (Figure 1a,b). The inset in Figure 1a is the HRTEM image of a single MoS_2_ QD with ordered lattice fringes. The lattice spacing is ~2.3 Å, which can be attributed to the (103) plane of crystalline MoS_2_. AFM was further used to check the thickness of the MoS_2_ QDs. Figure 1c also shows the monodispersity and size uniformity of the MoS_2_ QDs. Additionally, the profile shown in Figure 1d indicates that the height of MoS_2_ QDs is ~0.7 nm, demonstrating a monolayer nature [29]. The optical properties of absorption and photoluminescence spectra are presented in Figure 1e,f. A typical excitonic peak of MoS_2_ QDs at 310 nm with a remarkable shoulder is shown in Figure 1e [22]. The fluorescence emission peak is 430 nm under 350 nm excitation (Figure 1f). The photoluminescence quantum yield of MoS_2_ QDs was measured choosing quinine sulfate (54%, 350 nm excitation) as a standard sample. According to the data in Figure 1e,f, the photoluminescence quantum yield of MoS_2_ QDs is calculated at 20.4%, which is relatively higher than those reported previously for MoS_2_ QDs fabricated with a similar hydrothermal method [24,27]. In this study, the GSH was used as a reductant and passivation agent, which can provide enough surface functional groups to eliminate the edge defects, resulting in a high fluorescence emission.

The high-resolution XPS was conducted for further elementary analysis. The XPS response in the Mo 3d, S 2p is shown in Figure 2a,b, respectively. Figure 2a exhibits two peaks at 232.8 and 229.5 eV, which correspond to Mo^4+^ 3d_3/2_ and Mo^4+^ 3d_5/2_. Additionally, the two characteristic peaks of S 2p_1/2_ and S 2p_3/2_ located at 163.4 and 162.3 eV indicate a 2H phase for the crystal structure of MoS_2_ QDs [30,31]. Moreover, the atomic ratio of Mo/S is about 1:2, indicating the formation of MoS_2_ QDs. The formula of GSH is drawn in Figure 2c. During synthesis of MoS_2_ QDs, GSH was not only used as the reducing agent, but also as surface passivation agent to provide carboxyl, amino, and thiol groups for MoS_2_ QDs. The surface functional groups of MoS_2_ QDs were further verified by FTIR (Figure 2d). The characteristic peaks at 3212 and 3100 cm^−1^ (N-H stretching vibrations), 2600 cm^−1^ (S-H stretching vibration), 1684 cm^−1^ (C=O stretching vibration), 1590 cm^−1^ (N-H in-plane bending vibration), 1404 cm^−1^ (C-N stretching vibration of amide group), 1280 cm^−1^ (C-N stretching vibration of amine group), 1235 cm^−1^ (O-H stretching vibration), 1109 cm^−1^ (C-O stretching vibration) and 740 cm^−1^ (N-H out-plane bending vibration) indicate that carboxyl, amino and thiol groups of GSH can decorate the surface of MoS_2_ QDs to eliminate the edge defects, resulting in a high fluorescence emission.

The high fluorescence and environment-friendly characteristics of surface-functionalized MoS_2_ QDs make them a potential candidate for sensing metal ions. To explore the sensing capability of these MoS_2_ QDs, selectivity toward various metal ions was performed. Various metal ions with the same concentration of 25 μM were added into the MoS_2_ QDs aqueous solution to study the influence on respective MoS_2_ QDs fluorescence. The (F_0_ − F)/F_0_ value was used to determine the fluorescence enhancement or quenching of MoS_2_ QDs, where the fluorescence intensities of MoS_2_ QDs without/with metal ions were represented by F_0_ and F, respectively. Figure 3a indicates that only Pb^2+^ ions caused obvious fluorescence quenching, while the other metal ions have a slight impact on the fluorescence intensity. These results demonstrate that the surface-functionalized MoS_2_ QDs have high selectivity for Pb^2+^ ion.

To evaluate the stability of MoS_2_ QDs under different conditions, the effect of pH on the fluorescence of MoS_2_ QDs was also studied by adjusting the acidity to alkalinity. As shown in Figure 3b, the fluorescence intensity of MoS_2_ QDs shows a slight variation with increasing the pH in a wide range from 3.7 to 10.4, indicating negligible influence of pH on the fluorescence of MoS_2_ QDs. When the sample was kept for 50 days, the fluorescence intensity of MoS_2_ QDs was almost unchanged while the position of maximum emission peak was red shifted slightly (Figure 3c). These observations suggest that MoS_2_ QDs can be used as a stable fluorescence probe in complex underwater environments.

The sensitivity for Pb^2+^ ion detection was carried out by fluorescence titration of MoS_2_ QDs with varying concentration of Pb^2+^ ion from 0 to 120 μM. By increasing the concentration of Pb^2+^ ions from 0 to 120 μM, the fluorescence intensity of the MoS_2_ QDs decreases gradually (Figure 4a). Additionally, we plotted the degree of quenching (F_0_ − F)/F_0_ versus Pb^2+^ ion concentration in the range 0 to 60 μM (Figure 4b). We obtained a good linear relationship between (F_0_ − F)/F_0_ with Pb^2+^ ion concentration (R^2^ = 0.984), and the LOD was measured at 0.056 μM (3σ per slope, where σ is the standard deviation of blank signals, N = 10), which is less than the 15 μg/L safety value set by United States Environmental Protection Agency [32]. Moreover, compared with carbon dots and other inorganic probes used for the detection of Pb^2+^ by using fluorescence method (Table 1), the LOD obtained using MoS_2_ QDs as the sensor was slightly larger while the linear detection range was wider. However, compared with 1,4-diaminobutane (DAB) capped MoS_2_ QDs for monitoring the Pb^2+^ ions [21], the GSH-functionalized MoS_2_ QDs in this work possess higher quantum yield and lower LOD, which is suitable to use as a fluorescent probe for lead ions. These results suggest that the surface-functionalized MoS_2_ QDs can be used as a sensor for Pb^2+^ ion detection with high selectivity and sensitivity.

To further study the fluorescence quenching mechanism of MoS_2_ QDs for Pb^2+^ ion detection, we measured the photoluminescence decay dynamics of MoS_2_ QDs in the presence of Pb^2+^ ion (0, 5, 10, 15, 20 μM). The photoluminescence lifetime of MoS_2_ QDs decreases gradually with increasing concentration of Pb^2+^ ion (Figure 5a), indicating that Pb^2+^ ion plays an important role in the exciton recombination of MoS_2_ QDs. The photoluminescence decay curves in Figure 5a can be fitted with a bi-exponential model; the fitting parameters are shown in Table 2. There are two decay processes (short lifetime *τ*_1_ and long lifetime *τ*_2_) for the exciton deactivation. The fast component *τ*_1_ shows a mild variation while slow component *τ*_2_ becomes shorter with increasing concentration of Pb^2+^ ions. These two lifetime components could be assigned to band-edge emission and surface-state-assisted emission, respectively [21,36]. The shorter of slow component *τ*_2_ from surface-state-assisted emission may be ascribed to the observation that Pb^2+^ ion can chelate with surface functional groups of MoS_2_ QDs [33], resulting in electron transfer from the excited-state MoS_2_ QDs to the Pb^2+^ ion. The electron transfer rate *κ**_ET_* can be calculated by the following equation [37]:$$\kappa_{ET}=\frac{1}{\tau_{\mathrm{av}}}-\frac{1}{\tau_{0}},\;$$
where *τ*_av_ and *τ*_0_ are the average lifetime of MoS_2_ QDs with and without Pb^2+^ ions, respectively. When the Pb^2+^ ion concentration is 5 μM, the *κ**_ET_* is 3.26 × 10^7^ s^−1^. We further observed the linear dependence of electron transfer rate *κ**_ET_* on the Pb^2+^ ion concentration (Figure 5b). This evolution of *κ**_ET_* further indicates that the fluorescence quenching originates from nonradiative electron transfer from the excited MoS_2_ QDs to the Pb^2+^ ions.

## 4. Conclusions

We prepared high fluorescence MoS_2_ QDs by using an easily manipulated hydrothermal method, in which (NH_4_)_2_MoS_4_ and GSH were used as Mo and S sources. In this method, GSH was used as a reductant and capping agent to obtain high fluorescent MoS_2_ QDs. The morphological and structural characterization from TEM and AFM demonstrated that the ultrasmall MoS_2_ QDs (~4.4 nm) were monodispersed and single-layered. FTIR analysis further manifested that the functional groups provided by GSH can passivate the surface of MoS_2_ QDs. The photoluminescence of MoS_2_ QDs was quenched in the presence of Pb^2+^ ions. Based on the quenching effect, the surface-functionalized MoS_2_ QDs were used as a fluorescence probe with high sensitivity and selectivity to detect Pb^2+^ ions. From the fluorescence titration of MoS_2_ QDs, we determined the sensitivity of this sensor at the detection limit of 0.056 μM. Moreover, the photoluminescence decay dynamics of MoS_2_ QDs help us to understand the electron transfer behavior between MoS_2_ QDs and Pb^2+^ ions. A linear relationship between electron transfer rate *κ**_ET_* and Pb^2+^ ion concentration was observed. Our findings indicate that this water-soluble nanomaterial holds great promise and would be a good sensor for the detection of lead ions.

## Figures and Tables

**Figure 1 nanomaterials-12-03329-f001:**
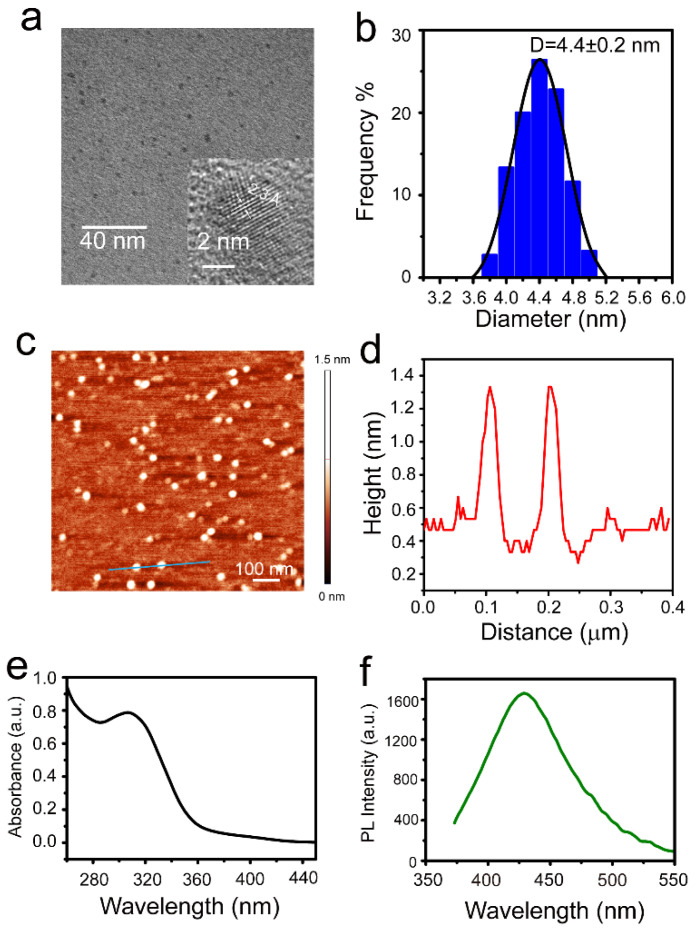
(**a**) TEM image of monodispersed MoS_2_ QDs, and the inset shows the HRTEM image with lattice spacing; (**b**) diameter distribution histogram of MoS_2_ QDs; (**c**) AFM image of MoS_2_ QDs; (**d**) corresponding height profile of blue line labeled in (**c**); (**e**) UV–vis absorption; (**f**) photoluminescence spectra of MoS_2_ QDs.

**Figure 2 nanomaterials-12-03329-f002:**
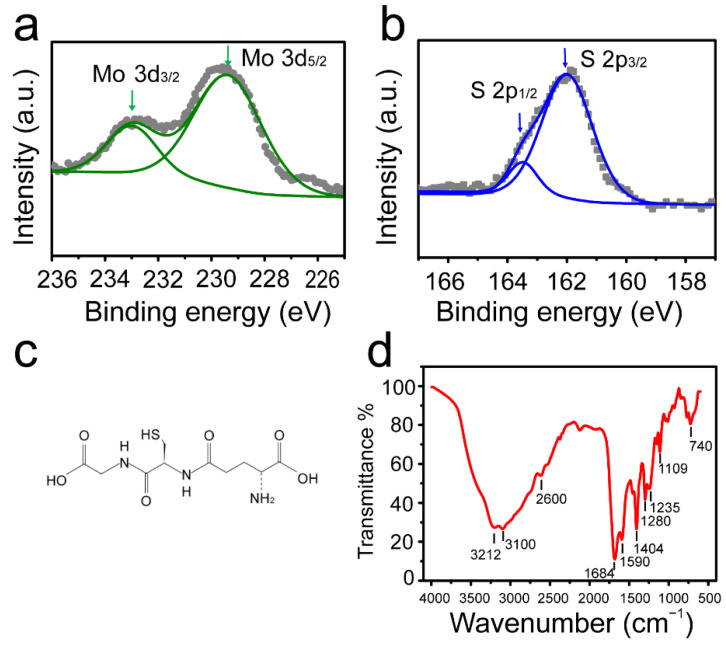
(**a**,**b**) Typical XPS response in Mo 3d and S 2p regions, and the solid line shows the fitting result of experimental data; (**c**) formula of GSH; (**d**) FTIR spectrum of surface functionalized MoS_2_ QDs with carboxyl, amino and thiol groups.

**Figure 3 nanomaterials-12-03329-f003:**
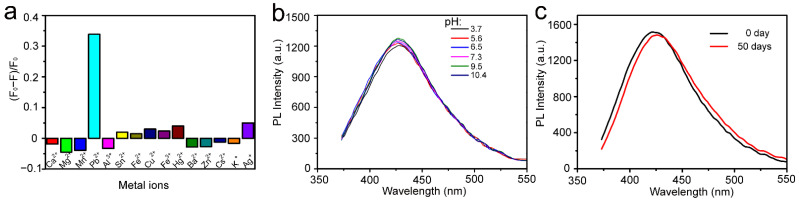
(**a**) The fluorescence enhancement or quenching of MoS_2_ QDs in the presence of various metal ions. (**b**) Photoluminescence spectra of MoS_2_ QDs under different pH conditions. (**c**) Photoluminescence spectra of MoS_2_ QDs stored at 4 °C for 0 and 50 days.

**Figure 4 nanomaterials-12-03329-f004:**
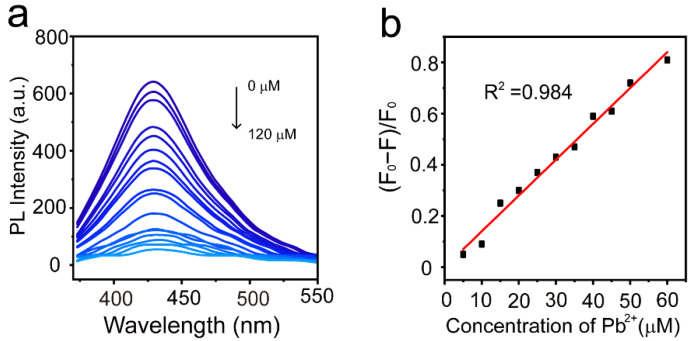
(**a**) Fluorescence spectra of MoS_2_ QDs with different concentrations of Pb^2+^ ion from 0 to 120 μM; (**b**) linear curve between (F_0_ − F)/F_0_ and Pb^2+^ ion concentration in the range 5 to 60 μM.

**Figure 5 nanomaterials-12-03329-f005:**
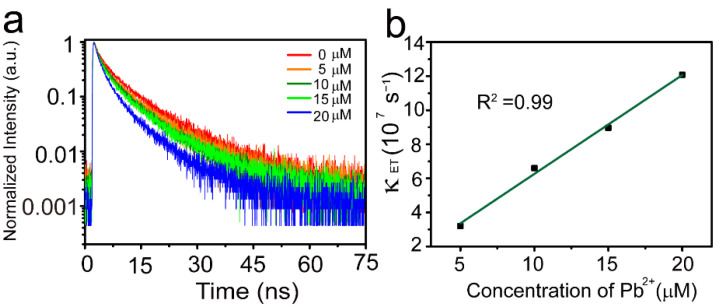
(**a**) Photoluminescence decay curves of MoS_2_ QDs in presence of different concentrations of Pb^2+^ ions (0, 5, 10, 15, 20 μM) under pulsed excitation at 350 nm. (**b**) Dependence of electron transfer rate on concentrations of Pb^2+^ ions.

**Table 1 nanomaterials-12-03329-t001:** Comparison of various fluorescent sensors for Pb^2+^ ion detection.

Sensors	Linear Range	LOD	Reference
Carbon dots	0.01–1 μM	0.59 nM	[10]
Carbon dots	0.033–1.67 μM	12.7 nM	[33]
ZnSeS/Cu:ZnS/ZnS QDs	0.04–6 μM	21 nM	[9]
Mn-doped ZnS QDs	1–100 μg/L	0.45 μg/L	[8]
Gold nanoparticles-DNAzyme	10–2500 nM	1.7 nM	[34]
N-CDs/R-CDs@ZIF-8	0.05–50 μM	4.78 nM	[35]
MoS_2_ QD_S_	33 μM–8 mM	50 μM	[21]
MoS_2_ QD_S_	5–60 μM	0.056 μM	This work

**Table 2 nanomaterials-12-03329-t002:** Bi-exponential fitting for photoluminescence decay curves of the MoS_2_ QDs. Fitting equation: *I*(t) = *I*_0_ + *A*_1_exp(−*t*/*τ*_1_) + *A*_2_exp(−*t*/*τ*_2_), and the average lifetime *τ*_av_ = (*A*_1_*τ*_1_^2^ + *A*_2_*τ*_2_^2^)/(*A*_1_*τ*_1_ + *A*_2_*τ*_2_), where *A*_1_, *A*_2_ and *τ*_1_, *τ*_2_ are the amplitudes and lifetimes, respectively.

Concentration	*A* _1_	*A* _2_	*τ*_1_ (ns)	*τ*_2_ (ns)	*τ*_av_ (ns)
0 μM	0.34	0.15	3.1	12.8	9.4
5 μM	0.37	0.09	3.1	11.6	7.2
10 μM	0.38	0.09	2.9	9.5	5.8
15 μM	0.38	0.08	2.8	8.6	5.1
20 μM	0.39	0.08	2.5	7.5	4.4

## Data Availability

The data that support the findings of this study are available from the corresponding author upon reasonable request.

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
