# Peer review of "One-Step Hydrothermal Synthesis of Highly Fluorescent MoS2 Quantum Dots for Lead Ion Detection in Aqueous Solutions"

_nanomaterials, 2022, doi:10.3390/nano12193329_

Round 1

Reviewer 1 Report

In the presented work entitled „One-step Hydrothermal Synthesis of High Fluorescent MoS2 Quantum Dots for Lead Ions Detection in Aqueous Solution” by L. Xie et al., the Authors synthesize MoS2 quantum dots via hydrothermal method and investigate their photoluminescence spectra. This is done to test detection effectiveness for the heavy metal ions (here for the Pb2+ ion). Based on their analysis, they observe that the photoluminescence quantum yield has room for improvement within the hydrothermal method. Moreover, the prepared quantum dots present high selectivity as well as sensitivity for Pb2+ ion and have great potential to become a good substitute for lead ion detection. 

The presented manuscript is written in a clear and well organized manner, while the analysis seems to be free of any major errors.  Therefore, I would like to recommend present paper for publication after the following minor corrections (which I believe should improve the paper even further). In particular:

1.     Some small editorial mistakes can be noticed. For examples, after Eq. 1 one would expect comma and the following sentence should not start with the capital letter “W”. In text Table 1 is described with the roman “I”, please unify. In line 213 and 214 the “ET” is not written as a lower index of kappa. Please review the paper for this mistakes and other similar and correct accordingly.

2.     When the Authors mention interesting electronic properties of MoS2 and WS2 it would be instructive to actually refer to some recent research. For example, please refer to their applications as the nanoscale field effect transistors (Small 125676-5683 (2016)) the low-dimensional rectifying contacts (Physical Review B 97, 195315 (2018)) the phototransistor devices (ACS Nano 13, 9638–9646 (2019)) the spin-valley filters (Physical Review B 101, 115423 (2020)) or the photodetectors (RSC Adv.,10, 30529-30602 (2020). This list can be briefly extended if the Authors would like to add some other interesting examples. Such extension of the discussion will surely improve the context for the paper.

3.     I cannot find proper discussion why the prepared quantum dots exhibits higher photoluminescence quantum yield than the quantum dots prepared using the same method in other recent studies.

4.     When making comparisons with other results or speaking of some limits, all the sources of such data should be properly cited. For example, reference to the limits of the United States Environmental Protection Agency is cleary missing in line 193. Please review and correct paper accordingly.

5.     To this end, I think that conclusions are just too short, simple and just a straightforward description of the obtained results. I would like to encourage the Authors to improve this section so the Readers can get some more new insight from their conclusions.

*end of report*

Author Response

Dear professor:

   We are very grateful to the referee for the insightful comments and helpful suggestions, specially thank you for the positive points on our manuscript. The related modification and data analysis have been added into the revised manuscript. For clarity, we generalized the comments and concerns into a few points and answered them accordingly.

Reviewer 2 Report

The authors report an hydrothermal method for the preparation of ghutathione-MoS2 QDs and their use as fluorescent probes for the detection of Pb2+ ions. The work is of interest and the results are relatively clearly presented. The following comments should be considered by the authors :

- along the whole manuscript, results should be better discussed in the context of literature. For example, MoS2 QDs have already been used for the detection of Pb2+ (see Mater. Res. Bull. 2020, 131, 110978). The reference was cited by the authors but not discussed.

-indicate if the PL intensity of GSH-capped MoS2 QDs is dependent on the pH of the solution.

- the authors must evaluate the colloidal stability as well as the photostability of GSH-capped MoS2 QDs.

- the authors must compare the performance of MoS2 QDs for the detection of Pb2+ to other QDs described in the literature (see for example, J. Photochem. Photobiol. A 2022, 431, 114050; Sensors and Actuators B 2017, 242, 679-686; Spectrochimica Acta A 2016, 164, 98-102; ...). The advances made should be highlighted.

Author Response

 Dear professor,

  We are very grateful to your insightful comments and helpful suggestions, specially thank you for the positive points on our manuscript. The related modification and data analysis have been added into the revised manuscript. For clarity, we generalized the comments and concerns into a few points and answered them accordingly.

Round 2

Reviewer 2 Report

Most of the corrections suggested were made by the authors. The following comment must still be considered:

- page 1, line 41-42 : the inorganic quantum dots cited by the authors (ZnS, ZnSeS) do not contain heavy metal cations. It would be interesting to compare the toxicity of Zn and Mo.

- the authors must compare the LODs of the various fluorescent inorganic probes used for the detection of Pb2+ and highlight the advances made using MoS2 quantum dots.

Author Response

We thank the referee for the helpful suggestions. Following the referee’s suggestion, we have revised the manuscript and added the comparison of various fluorescent sensors for Pb2+ ions detection in table 1 in the revised manuscript.
